# Serum and Urinary Matrix Metalloproteinase-9 Concentrations in Dehydrated Horses

**DOI:** 10.3390/ani13243776

**Published:** 2023-12-07

**Authors:** Julia N. van Spijk, Hsiao-Chien Lo, Roswitha Merle, Ina-Gabriele Richter, Anne Diemar, Sabita D. Stoeckle, Heidrun Gehlen

**Affiliations:** 1Equine Clinic, Free University of Berlin, 14163 Berlin, Germany; jvanspijk@vetclinics.uzh.ch (J.N.v.S.); cactusfhky8@gmail.com (H.-C.L.); heidrun.gehlen@fu-berlin.de (H.G.); 2Vetsuisse Faculty, Equine Department, University of Zurich, Winterthurerstrasse 260, 8057 Zurich, Switzerland; 3Institute for Veterinary Epidemiology and Biostatistics, Free University of Berlin, 14163 Berlin, Germany; roswitha.merle@fu-berlin.de; 4Research Centre of Medical Technology and Biotechnology, Department of Cell Biology, 99947 Bad Langensalza, Germany; irichter@fzmb.de (I.-G.R.); adiemar@fzmb.de (A.D.); 5Equine Clinic, University of Leipzig, An den Tierkliniken 11a, 04103 Leipzig, Germany

**Keywords:** matrix metalloproteinase, biomarker, renal function, dehydration, gelatinase B, equine kidney disease, MMP-9, dehydration

## Abstract

**Simple Summary:**

Kidney disease is difficult to diagnose since available laboratory parameters only increase later in time. Matrix metalloproteinase-9 is an enzyme found in renal tissue and increased in human kidney disease. This study evaluated serum MMP-9 (sMMP-9) and urinary MMP-9 (uMMP-9) concentrations in dehydrated horses. Horses with signs consistent with dehydration and four healthy clinic-owned horses were included. Blood and urinary samples for MMP-9 measurements were collected at admission, and after 12 h, 24 h, and 48 h. Furthermore, the serum creatinine, urea, symmetric dimethylarginine (SDMA), urine-specific gravity, urinary protein concentration, fractional sodium excretion, and urinary gamma–glutamyl transferase/creatinine ratio (uGGT/Cr) were measured. Statistical analysis included a repeated measures ANOVA and mixed linear regression model. Overall, 40 dehydrated horses were included (Acute kidney injury 1/40; elevated serum creatinine 7/40; elevated serum SDMA 11/40; elevated uGGT/Cr at presentation 5/28). In dehydrated horses, sMMP-9 concentrations were significantly higher on arrival compared to 12 h, 24 h, and 48 h. In healthy horses, sMMP-9 showed no differences over time or compared to patients. Urinary casts and uGGT/Cr correlated with sMMP-9. Serum urea was associated with uMMP-9/Cr. In conclusion, sMMP-9 was elevated at arrival in dehydrated patients compared to later measurements. Correlations to uGGT/Cr and urinary casts need further evaluation.

**Abstract:**

Matrix metalloproteinase-9 is increased in renal tissue in human kidney disease, but its role as a biomarker for kidney disease has not been fully evaluated yet. The aim of this study was to evaluate serum MMP-9 (sMMP-9) and urinary MMP-9 (uMMP-9) concentrations in dehydrated horses. Dehydrated horses were prospectively included. Blood and urinary samples were taken at admission, and after 12, 24, and 48 h (t0, t12, t24, t48), an anti-equine MMP-9 sandwich ELISA was used. Four healthy horses served as the controls. Serum creatinine, urea, symmetric dimethylarginine (SDMA), urine-specific gravity, urinary protein concentration, fractional sodium excretion, and urinary gamma–glutamyl transferase/creatinine ratio (uGGT/Cr) were measured. Statistical analysis included a repeated measures ANOVA and mixed linear regression model. Overall, 40 dehydrated horses were included (mild dehydration 13/40, moderate 16/40, severe 11/40). Acute kidney injury was found in 1/40 horses; 7/40 horses showed elevated serum creatinine, 11/40 horses elevated serum SDMA, and 5/28 elevated uGGT/Cr at presentation. In dehydrated horses, sMMP-9 concentrations were significantly higher on t0 (median: 589 ng/mL, range: 172–3597 ng/mL) compared to t12 (340 ng/mL, 132–1213 ng/mL), t24 (308 ng/mL, 162–1048 ng/mL), and t48 (258 ng/mL, 130–744 ng/mL). In healthy horses, sMMP-9 (239 ng/mL, 142–508 ng/mL) showed no differences over time or compared to patients. uMMP-9 and uMMP-9/creatinine did not differ over time or to the controls. No differences were found between dehydration groups. Urinary casts (*p* = 0.001; estimate = 135) and uGGT/Cr (*p* = 0.03; estimate = 6.5) correlated with sMMP-9. Serum urea was associated with uMMP-9/Cr (*p* = 0.01, estimate 0.9). In conclusion, sMMP-9 was elevated at arrival in dehydrated patients compared to later measurements. Correlations to uGGT/Cr and urinary casts need further evaluation.

## 1. Introduction

Early identification of renal disease is difficult due to the large reserve function capacity of the kidneys, leading to a delayed identification of renal function impairment since commonly measured renal parameters such as serum urea and creatinine concentrations only increase at a later stage [1]. Detection of renal damage, on the other hand, including markers such as urinary tubular enzyme concentrations, does not give any information on renal function [2]. Furthermore, definition of kidney injury in equine medicine is variable, and standardized definitions such as those used in humans are unavailable [3,4]. When using an adapted definition of kidney injury in horses, the prevalence of acute kidney injury in horses was recently described to be 15% in 325 equine emergency cases, mainly colic patients [5]. 

Matrix metalloproteinase-9 (MMP-9), or gelatinase B, is a complex multidomain enzyme which serves a multitude of physiological processes. In addition to its function to remodel the extra-cellular matrix, this enzyme cleaves other substrates and plays a role in inflammation, cell proliferation, migration, differentiation, and apoptosis [6]. Several matrix metalloproteinases are expressed in kidney tissue, and increased concentrations of MMP-9 were found in human kidney disease, such as in acute ischemic renal disease, diabetic nephropathy, and other less common nephropathies such as lupus nephritis or postinfectious glomerulonephritis [7]. 

Urinary MMP-9 concentrations correlate with urea, creatinine, glomerular filtrations rate, and proteinuria in renal disease in humans [8,9,10]. In children suffering from Henoch–Schönlein purpura nephritis, the urinary MMP-9 concentration was significantly higher than in healthy children, whereas the difference was not significant in the serum MMP-9 concentration [11]. However, until now, studies that evaluated the use of MMP-9 as a biomarker to detect renal disease are rare, and its use as such is largely unknown. 

Little is known about MMP-9 measurements in horses and its use as a marker of renal disease. Increased urinary MMP-9 (uMMP-9) concentrations were found in colic horses, and a role in equine kidney damage was proposed [12]. 

The aim of this study was to evaluate serum and urinary MMP-9 concentrations in dehydrated horses. We hypothesized that MMP-9 measurements would correlate with renal function and/or dehydration severity. 

## 2. Material and Methods

Clinically dehydrated adult horses presented between August 2018 and December 2019 at the equine clinic of the Free University of Berlin were included in a prospective study. The study protocol is also described elsewhere [13]. In brief, horses were grouped into three groups of dehydration (mild dehydration 6–8% body weight (bwt), moderate dehydration 8–10% bwt, severe dehydration > 10% bwt) based on clinical examination, PCV, and TP on admission (Table 1). If not all criteria were consistently matched with one group of dehydration, the horse was allocated to the group to which most of the criteria fit.

Serum blood samples were taken at admission and after 12, 24, and 48 h (t0, t12, t24, and t48, respectively). Urine samples were taken within two hours at the same time points either by free catch or catheterization. The urinary samples were stored at 4–8 °C until urinalysis.

Four healthy teaching mares belonging to the Free University of Berlin were included as a control group. Horses were considered healthy from a clinical exam and normal blood work including hematology and chemistry profile. Serum samples were taken four times: at start point (t0), and 12, 24, and 48 h later (t12, t24, t48) by venipuncture, and a single urine sample was taken by free catch or urinary catheter within the first two hours.

Serum creatinine and urea were measured using an automated AU680 clinical chemistry analyzer (Beckmann Coulter GmbH, Krefeld/Germany). Serum symmetric dimethylarginine (SDMA) was measured with a DLD SDMA ELISA Kit previously used in horses (DLD Diagnostik GmbH, Hamburg/Germany) [13,14].

Urine samples were analyzed by a dip stick Combur 9 Test (Roche Deutschland Holding GmbH, Grenzach-Wyhlen/Germany) immediately and sent to an external laboratory (SYNLAB vet GmbH, Augsburg/Germany) for measurement of USG, urinary protein concentration, fractional sodium excretion, and uGGT/Cr using the AU680 clinical chemistry analyzer (Beckman Coulter GmbH, Krefeld/Germany) and for microscopic sediment analysis within 24 h.

Serum and urine samples were frozen at −80 °C until all samples were collected, and MMP-9 measurements were then performed as a batch in June 2021 (serum samples) and October 2021 (urine samples). The MMP-9 ELISA used in this study was used successfully in previous equine studies [15,16]. Measurements were performed at the Research Centre of Medical Technology and Biotechnology, Department of Cell Biology, Bad Langensalza, Germany. Each sample was measured three times in duplicates, and the mean value of these six measurements per sample was calculated. The concentration of MMP-9 was measured using an anti-equine MMP-9 sandwich ELISA (equine MMP-9 FAST ELISA Kit, fzmb GmbH, Bad Langensalza, Germany). The microtiter plate provided in this kit was pre-coated with an antibody specific to equine MMP-9. To each well was added 100 μL of the sample and parallel 100 μL of each standard solution containing 24 ng/mL, 12 ng/mL, 6 ng/mL, 4 ng/mL, 2 ng/mL, 1 ng/mL, 0.5 ng/mL, and 0.25 ng/mL equine MMP-9, and they were incubated for 30 min at 37 °C. The wells were then rinsed 4 times with Working Wash Solutions (200 μL per well), and 100 μL anti-equine MMP-9 horse radish peroxidase (HRP) conjugate was added. This was again incubated for 30 min at 37 °C before the wells were rinsed 5 times with Working Wash Solution (200 μL per well), and 100 μL of tetramethylbenzidine (TMB) substrate was added and incubated for 10 min at room temperature in the dark. The enzyme-substrate reaction was terminated by the addition of 100 μL of a 1.2 M sulphuric acid solution. Immediately afterwards, the color change was measured spectrophotometrically at a wavelength of 450 nm ± 10 nm. For the creation of the standard curve, the measured optical density was plotted against the known concentration of each calibration standard, whereby the curve was fitted using a 5-parameter logistic (5PL) model. Based on the established 5PL equation, the MMP-9 concentration of the samples was then calculated using the measured optical density of each sample. Referring to the standard curve, the quantification range was 0–900 ng/mL in serum samples and 0.15–120 ng/mL in urine samples. The MMP-9 ELISA used was validated against zymography and has an intra-assay coefficient of variation (CV) < 9% as well as an inter-assay CV < 15% according to the manufacturer’s information.

IBM SPSS version 27 was used to analyze the data. Data are presented as median and range due to low numbers. A repeated measures ANOVA with Bonferroni post hoc correction was used to analyze the differences in sMMP-9, uMMP-9, and uMMP-9/Cr between the timepoints and groups. A mixed linear regression model was used to evaluate the effect of creatinine, urea, SDMA, USG, urinary protein, urinary GGT/creatinine ratio, and the presence of urinary casts (fixed factors) on sMMP-9, uMMP-9, and uMMP-9/Cr (dependent variables). Horse was set as a random factor. The level of significance was set at *p* < 0.05.

## 3. Results

Samples of 40 dehydrated horses were included in this study. Most horses (30/40, 75%) were diagnosed with gastrointestinal diseases, including strangulating small intestinal lesions (10/30, 33%), enteritis/colitis (5/30, 16.7%), meteorism, and displacement of the colon, including torsio coli (6/30; 20%), large intestinal impaction (5/30, 16.7%), small intestinal impaction (1/30, 3.3%), gastric impaction (1/30, 3.3%), peritonitis (1/30, 3.3%), and colic of unknown cause (1/30, 3.3%). Less common were orthopedic diseases (laminitis and fracture patients, each 1/40, 2.5%), hemoabdomen of unknown cause (1/40, 2.5%), dystocia (1/40 2.5%), and unknown diagnosis (5/40, 12.5%). Breeds included Warmbloods (11/40, 28%), Ponies (11/40, 27.5%), Thoroughbreds (6/40, 15%), Draft horses (3/40, 7.5%), others (8/40, 20%), and unknown breed (2/40, 5%). Age ranged from 4 months to 28 years (median: 15.5 years); 19/40 (47.5%) were geldings and 21/40 (52.5%) were mares. Urinary samples could not be collected for all of the horses and at all time points since many of the horses did not urinate spontaneously and catheterization was not indicated from a clinical point of view. In total, 28 urinary samples were available, of which 23 were not only suitable for laboratory measurements but also for sediment analysis.

Upon admission to the clinic (t0), 13/40 (32.5%) horses were classified as mildly (6–8% bwt), 16/40 (40%) as moderately (8–10% bwt), and 11/40 (27.5%) as severely dehydrated (>10% bwt).

Elevated serum creatinine values (>167 μmol/L) were found in 7/40 (18%) horses at t0, 3/40 (7.5%) at t12, 2/40 (5%) at t24, and 1/40 (2.5%) at t48. A single horse developed acute kidney injury defined as an increase ≥26.5 μmol/L within 48 h [5].

Elevated serum SDMA was present in 11/40 (27.5%) horses (11/40 on t0, 2/40 on t12, 5/40 on t24, 3/40 on t48). Elevated uGGT/Cr occurred in 5/28 (17.8%) horses (5/28 on t0, 2/28 (7.1%) on t12, and 1/28 (3.6%) on t24 and t48). Urinary casts were present in 16/23 (7%) horses on t0, in 2/13 (15%) on t12, in 4/11 (36%) on t24, and 2/10 (40%) on t48.

Serum MMP-9 concentrations in dehydrated horses were significantly higher on t0 (median: 589 ng/mL, range: 172–3597 ng/mL) compared to t12 (median: 340 ng/mL, range: 132–1213 ng/mL), t24 (median: 308 ng/mL, range: 162–1048 ng/mL), and t48 (median: 258 ng/mL, range: 130–744 ng/mL; *p* < 0.001; Figure 1). In healthy horses, sMMP-9 did not differ over time, and the median was 249 ng/mL (range: 209–303 ng/mL) at t0, at t12 the median was 227 ng/mL (range: 142–385 ng/mL), at t24 the median was 212 ng/mL (range: 165–480 ng/mL), and at t 48 the median was 238 ng/mL (range: 211–508 ng/mL). There was no statistically significant difference in sMMP-9 between healthy and dehydrated horses at any timepoint.

Urinary MMP-9 concentrations were 1.2 ng/mL (0.2–120 ng/mL) at t0, 0.6 ng/mL (0.2–3.7 ng/mL) at t12, 0.8 ng/mL (0.2–3.9 ng/mL) at t24, and 0.7 ng/mL (0.2–3.1 ng/mL) at t48. Urinary MMP-9 concentrations did not significantly differ over time or compared to healthy horses (0.6 ng/mL, range 0.2–1.2 ng/mL at t0). Urinary MMP-9/creatinine ratios did not significantly differ over time and were 0.6 ng/mL (0.0–39.5 ng/mL) at t0, 1.3 ng/mL (0.4–12.1 ng/mL) at t12, 0.7 ng/mL (0.5–6.7 ng/mL) at t24, and 0.7 ng/mL (0.0–7.0 ng/mL) at t48 in dehydrated horses and 0.9 ng/mL (0.0–39.5 ng/mL) at t0 in the healthy controls.

No differences in the sMMP-9, uMMP-9, or uMMP-9/Cr concentrations were found between dehydration groups (Figure 2).

The MMP-9 measurement results including mean, variation, standard deviation, and standard error of the mean of all six measurements of each sample can be found in Appendix A (serum samples) and Appendix A (urine samples).

Urinary casts (*p* = 0.001; estimate = 135, CI 55–216) and uGGTCr (*p* = 0.03; estimate = 6.5, CI 0.7–12.3) were statistically significantly associated with sMMP-9 measurements. Serum urea was statistically significantly associated with uMMP-9/Cr (*p* = 0.01, estimate 0.9, CI 0.2–1.6). The variance between horses was high in both analyses (79% in sMMP-9, 97% in uMMP-9/Cr), indicating a large effect of the individual horses. None of the variables was statistically significantly associated with uMMP9.

## 4. Discussion

This study reports serum and urinary MMP-9 measurements in 40 dehydrated horses. A statistically significant increase in sMMP-9 was found on arrival compared to t12, t24, and t48 in dehydrated horses. Additionally, correlations between the presence of urinary casts and uGGT/Cr with sMMP-9 concentrations and a correlation of serum urea with uMMP-9/Cr were found. However, since the urinalysis (especially the analysis of the sediment) could not be performed immediately after urine collection, some horses with acute kidney injury might have been missed. Furthermore, another limit of the study is that the progress of rehydration was not recorded. The correlation of the MMP-9 measurements and the resolution of the clinical evidence of dehydration would be useful in this evaluation.

In a previous study evaluating MMP-9 and MMP-2 measurements in five colic horses, the plasma values of MMP-2 were statistically significantly higher in sick horses, while the plasma MMP-9 values were higher, but not statistically significantly increased [12]. Unfortunately, absolute values cannot be compared between these two studies as different methods were used to measure MMPs. Most horses in our study were colic horses and therefore included a similar population of sick horses. We report serum MMP-9 measurements in a larger number of horses and were able to show a statistically significant difference in sick horses on arrival compared to later measurements. The serum MMP-9 measurements at arrival were also higher than in control horses; however, overall, there was no statistically significant difference found between dehydrated and control horses.

The plasma MMP-9 measurements were shown to increase after lipopolysaccharide infusion in horses, and peritoneal MMP-9 concentrations were shown to be a valuable biomarker to assess endotoxemia in equine colic [16,17]. The shown increase in sick, dehydrated colic horses might therefore also be a result of ongoing endotoxemia rather than an indicator of renal insufficiency. Interestingly, the serum MMP-9 measurements were, however, correlated with markers of renal tubular damage such as the presence of urinary casts and uGGT/Cr measurements. Whether there is a direct causative correlation or rather an indirect effect on endotoxemic horses to concurrently show renal tubular damage is unknown. Nevertheless, sMMP-9 might act as an indicator of renal damage in these patients.

The urinary excretion of MMP-9 is considered to be a potential indicator of renal tubular damage as the molecule is not filtered through an intact glomerulus [12,18]. Increased concentrations of MMP-9 were found in human kidney tissue in different diseases, such as in ischemic lesions or diabetic nephropathy [7]. Urinary proMMP-9 and MMP-9 concentrations were increased in equine colic cases; however, only the first was statistically significant in a small study including five horses [12]. In our data, we were not able to detect any significant changes in uMMP-9 concentrations between groups or over time. This might be influenced by low numbers, especially in control horses where urinary samples were only taken at one timepoint. Urinary excretion is dependent on renal blood flow and excretion rate, and this might influence urinary measurements. Therefore, creatinine values are commonly used as an internal reference, and we included uMMP9/Cr ratios to eliminate an eventual effect of renal blood flow, which is likely to be reduced in dehydrated patients. There was a mild correlation seen between uMMP9/Cr and serum urea, but the usefulness of uMMP-9 and/or uMMP-9Cr measurements to indicate renal function is unclear and needs to be further evaluated. Furthermore, it is possible that the two-hour time frame between blood sampling and urine collection may have contributed to differences in the hydration status of the patient since immediate treatment was indicated and pursued, so that the level of dehydration at the time of blood collection and urine collection may not match. This could partially explain the lack of correlation between urinary and serum MMP-9 concentrations in this study.

To our knowledge, the equine MMP-9 FAST ELISA Kit we used is the only one which uses native equine MMP-9 as standard compared to other commercial ELISA kits which predominantly use recombinant protein as standard. Moreover, the antibodies included in the kit are equine-specific, and native equine MMP-9 served as the immunogen for their generation. Most commercial equine MMP-9 ELISA kits use antibodies directed against recombinant equine MMP-9, such as those produced in *E. coli*. This could be detrimental due to the glycosylation of MMP-9 since *E. coli* lacks protein glycosylation, which can negatively affect both the protein folding of the recombinant protein and antigen recognition of the native protein in equine samples [19]. Taken together, the MMP-9 ELISA used in this study provides the best reliable results previously confirmed by the zymography analysis of synovial fluids from horses with degenerative joint disease [15].

Limitations of this study include a small sample size and the clinical nature of the study, which lead to missing samples. Classification of dehydration was based on clinical examination and basic laboratory measurements and therefore might have misclassified horses. This population was chosen to detect the early stages of acute kidney injury as dehydration is a common cause of reversible azotemia and considered a common risk factor for the development of acute kidney injury [1]. To further evaluate the value of MMP-9 measurements as a biomarker to detect kidney dysfunction, studies in other patient populations such as in horses with other causes and forms of acute or chronic renal disease are warranted. The effect of ongoing inflammatory processes and endotoxemia need to be further evaluated as this may be a major influencing factor on MMP-9 measurements.

## 5. Conclusions

In conclusion, sMMP-9 was elevated at arrival in dehydrated patients, which might indicate reduced renal function or might have been influenced by other factors such as inflammatory processes or endotoxemia in these diseased horses. Correlations to uGGT/Cr and urinary casts need further evaluation as MMP-9 could potentially serve as an indicator of renal injury in horses. An equine-specific MMP-9 FAST ELISA Kit was successfully used for the detection of MMP-9 in serum and urine samples.

## Figures and Tables

**Figure 1 animals-13-03776-f001:**
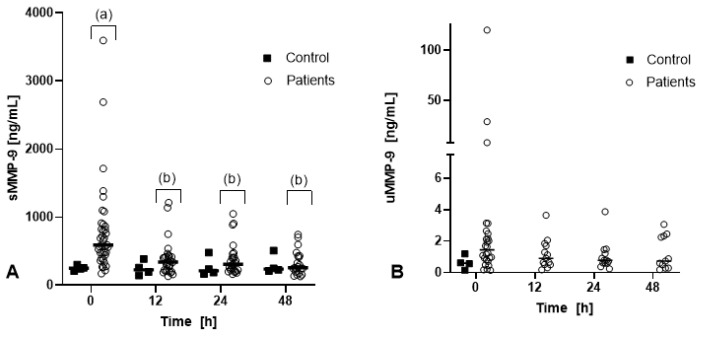
Serum (**A**) and urinary (**B**) MMP-9 concentration in 40 dehydrated horses (patients) and 4 control horses. Values shown are the means of six measurements of each sample. The median of all samples is indicated by a horizontal line. Different letters show statistically significant differences over time.

**Figure 2 animals-13-03776-f002:**
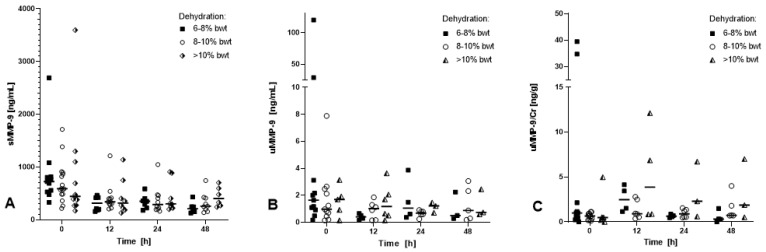
Serum MMP-9 (**A**), urinary MMP-9 (**B**), and urinary MMP-9/creatinine ratio (**C**) measurements in 40 horses with mild (6–8% bwt), moderate (8–10% bwt), and severe (>10%) dehydration. Individual values are shown; median is indicated by a horizontal line.

**Table 1 animals-13-03776-t001:** Grading of dehydration in horses.

Dehydration Grade [% Body Weight]	Heart Rate [/min]	Capillary Refill Time [s]	Packed Cell Volume [%]	Total Solids [g/L]
Mild (8% bwt)	40–60	2	40	70
Moderate (8–10% bwt)	61–80	3	45	75
Severe (>10% bwt)	>81	≥4	≥50	≥80

## Data Availability

Further data are available from the authors.

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
