# Peer review of "Serum and Urinary Matrix Metalloproteinase-9 Concentrations in Dehydrated Horses"

_animals, 2023, doi:10.3390/ani13243776_

Round 1

Reviewer 1 Report

Comments and Suggestions for Authors

The paper aims to evaluate the application of MMP-9 assays to determine renal disease or damage during dehydration due to a range if acute pathologies.  A technique to detect renal damage during acute clinical cases is sorely needed.  The careful evaluation of the application of MMP-9 has been attempted and generally well done. 

The major weakness is that there has been no direct determination of kidney damage in the cases, only inferred by clinical pathology and assumed as a sequelae or cause of dehydration.  Perhaps in the Introduction or Discussion the rationale for not more directly determining kidney damage e.g using the criteria in [5]?

63: re-word "late increase" - "delayed application"?  

83: "Little" rather than "few"

95: Supplementary 1 doesn't give ranges for CRT, PCV and TS for mold and moderate.  It is not indicated what was done if all criteria were not consistently matched.

255: the reasons for the missed samples should be described.

Comments on the Quality of English Language

Minor issues to be resolved.

Author Response

Dear reviewer, 

thank you very much for the review of our paper in order to improve its quality. 

Please find your comments and our answers below. 

The major weakness is that there has been no direct determination of kidney damage in the cases, only inferred by clinical pathology and assumed as a sequelae or cause of dehydration.  Perhaps in the Introduction or Discussion the rationale for not more directly determining kidney damage e.g using the criteria in [5]?

We appreciate your suggestion, but the mentioned paper included multiple measurements of the renal parameters during hospitalisation whereas our paper only included the first 48 hour period after admission (during this time period dehydration should be resolved).

63: re-word "late increase" - "delayed application"?  

Sentence completely rephrased.

83: "Little" rather than "few"

Done.

95: Supplementary 1 doesn't give ranges for CRT, PCV and TS for mild and moderate.  It is not indicated what was done if all criteria were not consistently matched.

For a better understanding, “supplementary” 1 was included as “table 1” in the text. Missing fields were filled in.

255: the reasons for the missed samples should be described.

In this study, samples were only taken if there was a clinical indication and were used accordingly for the study (blood tests, urinary catheter). If there was no clinical indication to insert a urinary catheter, spontaneous urine had to be collected. This was not possible in all cases, hence the reduced sample numbers.This information was included in the text.

Reviewer 2 Report

Comments and Suggestions for Authors

Manuscript Review

Serum and urinary matrix metalloproteinase-9 (MMP-9) concentrations in dehydrated horses.

Line 18: Provide reference regarding human MMP in kidney disease.

Line 38: Provide reference regarding human MMP in kidney disease.

Line 76: Please specific urine vs. serum for MMP increases in human kidney disease.

Line 93: Please clarify how many criteria were required to classify horse’s percent dehydration as this does not appear to be described in the referenced article. Heart rate alone, for example could vary widely based on degree of discomfort and may not represent dehydration well.

Line 165: Please expand upon the change in numbers of horses from x/40 which were clinically dehydrated to x/28 uGGT/Cr and x/23 urinary casts.

Line 229: I am not sure it is fair to call presence of MMP9 in the urine evidence of tubular damage. Is there evidence for increased secretion of MMP into tubules in the face of damage? If the molecule may be filtered after glomerulofiltration barrier breakdown, we would have to consider this evidence for glomerular damage. Consider broadening this statement.

Other Notes:

Generally, there is concern that urinary casts present in urine samples breakdown quickly after collection and that sediment analysis should be performed within 30 minutes of collection for optimal detection. Were these samples stored in a way to prevent sediment breakdown prior to analysis?

Do you have records of progress towards euvolemia in these patients – presumably many were in the process of being rehydrated. Correlation of resolution of clinical evidence of dehydration would be useful in this evaluation.

Consider that urine specific gravity changes rapidly in the face of changes to hydration status. Recognizing the challenges of collecting urine in a timely, noninvasive fashion in clinical patients, it is possible that a 2-hour window of urine collection to blood sampling may represent a fair amount of room for change in the patient’s hydration status, so the level of dehydration at the time of blood collection and urine collection may not match. This could partially explain the lack of correlation in this study. Consider including this as a limitation.

Comments on the Quality of English Language

Line 19: Space missing after horses.

Line 83: Replace “Few” with “Little”

Line 113: Replace “than” with “then”

Author Response

Dear reviewer

thank you very much for reviewing our paper in order to improve its quality. 

Please find your comments and our answers below. 

Serum and urinary matrix metalloproteinase-9 (MMP-9) concentrations in dehydrated horses.

Line 18: Provide reference regarding human MMP in kidney disease.

Done.

Line 38: Provide reference regarding human MMP in kidney disease.

Done.

Line 76: Please specific urine vs. serum for MMP increases in human kidney disease.

A recent study comparing serum and urinary MMP-9 in children was included as reference directly comparing serum and urinary measurements of MMP-9.

Line 93: Please clarify how many criteria were required to classify horse’s percent dehydration as this does not appear to be described in the referenced article. Heart rate alone, for example could vary widely based on degree of discomfort and may not represent dehydration well.

We have included the “supplementary 1” as “table 1” in the text to make it more understandable. If the criteria were partly fulfilled by more than one group, horses were allocated to the group for which they fulfilled most criteria.

Line 165: Please expand upon the change in numbers of horses from x/40 which were clinically dehydrated to x/28 uGGT/Cr and x/23 urinary casts.

In this study, samples were only taken if there was a clinical indication and were used accordingly for the study (blood tests, urinary catheter). If there was no clinical indication to insert a urinary catheter, spontaneous urine had to be collected. This was not possible in all cases, hence the reduced sample numbers.This information was included in the text.

Line 229: I am not sure it is fair to call presence of MMP9 in the urine evidence of tubular damage. Is there evidence for increased secretion of MMP into tubules in the face of damage? If the molecule may be filtered after glomerulofiltration barrier breakdown, we would have to consider this evidence for glomerular damage. Consider broadening this statement.

Changed renal tubular damage to renal damage.

Other Notes:

Generally, there is concern that urinary casts present in urine samples breakdown quickly after collection and that sediment analysis should be performed within 30 minutes of collection for optimal detection. Were these samples stored in a way to prevent sediment breakdown prior to analysis?

Unfortunately, the samples were not treated for sediment preservation. It is therefore possible, that some horses with AKI might have been undetected. We have included this in the manuscript.

Do you have records of progress towards euvolemia in these patients – presumably many were in the process of being rehydrated. Correlation of resolution of clinical evidence of dehydration would be useful in this evaluation.

Unfortunately, this information was not included in the evaluation. We have included this as a limitation in the text.

Consider that urine specific gravity changes rapidly in the face of changes to hydration status. Recognizing the challenges of collecting urine in a timely, noninvasive fashion in clinical patients, it is possible that a 2-hour window of urine collection to blood sampling may represent a fair amount of room for change in the patient’s hydration status, so the level of dehydration at the time of blood collection and urine collection may not match. This could partially explain the lack of correlation in this study. Consider including this as a limitation.

Included as suggested.

Reviewer 3 Report

Comments and Suggestions for Authors

Overall, the paper is well written and of interest for the readers. Some minor concerns are present.

mat and met:

please add some literature supporting the tests used or add an internal validation.

line 141: please add the post hoc test.

results: line 148-152: pease add prevalence as you did below.

line 157: please add a space between number and years.

line 159: please add prevlence.

lines 161-162: please put the prevalence at T12 etc.

lines 164-167: prevalences are missing.

line 169: at T0.

please add the word "median and range" in all the brackets.

please put the p value after higher in line 168.

please put a space (line 172).

please put the median values, you add only the range (see previous results how you wrote).

line 180-183: splease add median (word) and range.

Author Response

Dear reviewer

thank you very much for reviewing our paper in order to improve its quality. 

Please find your comments and our answers below. 

 mat and met:

please add some literature supporting the tests used or add an internal validation.

Included in the text.

line 141: please add the post hoc test.

Done.

results: line 148-152: please add prevalence as you did below.

Included in the text.

line 157: please add a space between number and years.

Included in the text.

line 159: please add prevlence.

Included in the text.

lines 161-162: please put the prevalence at T12 etc.

Included in the text.

lines 164-167: prevalences are missing.

Included in the text.

line 169: at T0.

Included in the text.

please add the word "median and range" in all the brackets.

Included in the text.

please put the p value after higher in line 168.

There was no statistical analysis performed since this only describes that one horse had acute kidney injury.

please put a space (line 172).

Included in the text.

please put the median values, you add only the range (see previous results how you wrote).

The median is before the brackets, the word was included in the text.

line 180-183: please add median (word) and range.

Included in the text.